# Toxicological and Safety Pharmacological Profiling of the Anti-Infective and Anti-Inflammatory Peptide Pep19-2.5

**DOI:** 10.3390/microorganisms10122412

**Published:** 2022-12-06

**Authors:** Clemens Möller, Lena Heinbockel, Patrick Garidel, Thomas Gutsmann, Karl Mauss, Günther Weindl, Satoshi Fukuoka, Dominik Loser, Timm Danker, Klaus Brandenburg

**Affiliations:** 1Life Sciences Faculty, Albstadt-Sigmaringen University, 72488 Sigmaringen, Germany; 2Brandenburg Antiinfektiva GmbH, c/o Forschungszentrum Borstel, 23845 Borstel, Germany; 3Biophysikalische Chemie, Martin-Luther-Universität Halle-Wittenberg, 06120 Halle (Saale), Germany; 4FG Biophysik, Leibniz Lungenzentrum, Forschungszentrum Borstel, 23845 Borstel, Germany; 5Sylter Klinik, Dr.-Nicolas-Straße 3, 25980 Westerland (Sylt), Germany; 6Abteilung Pharmakologie und Toxikologie, Pharmazeutisches Institut, Universität Bonn, 53121 Bonn, Germany; 7NMI-TT GmbH, 72770 Reutlingen, Germany

**Keywords:** sepsis, antiinflammatory, antiinfective, antimicrobial peptide

## Abstract

Aspidasept (Pep19-2.5) and its derivative Pep19-4LF (“Aspidasept II”) are anti-infective and anti-inflammatory synthetic polypeptides currently in development for application against a variety of moderate to severe bacterial infections that could lead to systemic inflammation, as in the case of severe sepsis and septic shock, as well as application to non-systemic diseases in the case of skin and soft tissue infections (SSTI). In the present study, Aspidasept and Aspidasept II and their part structures were analysed with respect to their toxic behavior in different established models against a variety of relevant cells, and in electrophysiological experiments targeting the hERG channel according to ICH S7B. Furthermore, the effects in mouse models of neurobiological behavior and the local lymph node according to OECD test guideline 429 were investigated, as well as a rat model of repeated dose toxicology according to ICH M3. The data provide conclusive information about potential toxic effects, thus specifying a therapeutic window for the application of the peptides. Therefore, these data allow us to define Aspidasept concentrations for their use in clinical studies as parenteral application.

## 1. Introduction

Bacterial infections are known to cause severe health-threatening conditions. Inflammation caused by antibiotic-resistant bacteria and sepsis constitute one of the most threatening medical challenges of our times [1]. One promising new therapeutic approach is the use of antimicrobial peptides (AMP), which may be based on naturally occurring compounds, such as defensins and cathelicidins, or which are newly designed based on effectively targeting the bacterial cell envelope and considering toxicological requirements. The latter approach was chosen by us, starting from the lipopolysaccharide (LPS) binding domain of the *Limulus* anti-LPS factor (LALF) and adapting the N- and C-terminal regions of the peptides with polar and non-polar amino acids, respectively, in an iterative way, until optimization was achieved with respect to binding to the lipid A moiety of LPS, the ‘endotoxic principle’ of LPS [2,3,4]. Using this development strategy/approach, it was possible to neutralize the inflammation elicited by LPS in in vitro systems, such as human mononuclear cells, as well as in animal models of sepsis. The lead structure Aspidasept^®^ (Pep19-2.5) and its derivative Pep19-4LF have a high binding affinity to LPS, with a binding constant of 2.8 · 10^8^/mol for Pep19-2.5 [5]. Both peptides show high endotoxin neutralization efficiency in vitro, and their antiseptic, as well as anti-inflammatory, effects have been demonstrated in vivo in mouse models of endotoxemia, bacteremia, and cecal ligation and puncture, as well as in an ex vivo model of human lung tissue [4,6]. It was found that gram-positive toxins and lipoproteins/-peptides were similarly neutralized as LPS also [7]. The broad neutralization efficiency and the additive action of the peptides with common antibiotics make them promising novel drug candidates against inflammation and bacterial sepsis [8,9,10]. Essential findings for the peptides are: (i) they inhibit the bacterial pathogen-associated molecular patterns (PAMPs) as constituents of the bacterial membrane, as well as in free form, (ii) they also act intracellularly [10], and (iii) correspondingly, they act against DAMPs (danger-associated molecular patterns), such as heparan sulfate [8].

A main prerequisite for a clinical trial authorization (CTA) is the safety evaluation of the drug candidate. For this, we have analysed the effects of the peptides in different established cell-based in vitro test systems. Further points are non-clinical safety studies and safety pharmacology according to ICH M3 (R2) (Step 5) and S7A (Step 5), which were performed in rats in repeated dose experiments with continuous intravenous application. In addition, safety tests in mice for (i) the determination of the influence of the drug on the central nervous system (modified Irwin test, test on neurobehavior) and (ii) the influence on the skin (skin sensitization) in the LLNA (local lymph node assay) were performed. Furthermore, tested also was a possible blockade of the cardiac hERG (human Ether-à-go-go-Related Gene) ion channel, which is correlated with life-threatening ‘torsades de pointes’ (cardiac arrhythmia) and has become a serious safety concern in drug discovery and development [11,12,13]. 

The current data now give a broad toxicological profile of Aspidasept^®^ with concentrations of 20 to 30 µg/mL in conditions where there are side effects observed in cellular systems; a NOAEL (No Observed Adverse Effect Level) and MTD (Maximum Tolerated Dose) of more than 3 mg/d∙kg and 20 mg/d∙kg, respectively, in rats in repeated toxicology tests with continuous intravenous application; a slight neurobehavioral effect at the highest dose (2 mg/kg, formulated in 0.9% saline); no effect on skin sensitization up to 5% formulated in pluronic aqueous solutions; and no effect on the hERG channel at concentrations well above 10 µg/mL. 

## 2. Materials and Methods

### 2.1. Peptides

Peptide synthesis: Peptides were synthesized in the Forschungszentrum Borstel with an amidated C terminus by the solid-phase peptide synthesis technique in an automatic peptide synthesizer (model 433A; Applied Biosystems, Seevetal, Germany) on Fmoc-Rink amide resin, according to the 0.1-mmol FastMoc synthesis protocol of the manufacturer, including the removal of the N-terminal Fmoc group. The peptidyl resin was deprotected and cleaved with a mixture of 90% trifluoroacetic acid (TFA), 5% anisole, 2% thioanisole, and 3% dithiothreitol for 90 min at room temperature. After cleavage, the suspension was filtered through a syringe filter into ice-cold diethyl ether. The precipitated peptides were separated by centrifugation and repeatedly washed with cold ether. The final purification was done by reverse-phase high-pressure liquid chromatography (RP-HPLC). Purity levels of up to 98% were achieved by using an Aqua C18 column (Phenomenex, Aschaffenburg, Germany) in combination with dedicated gradients of acetonitrile in 0.1% TFA, checked by matrix-assisted laser desorption ionization time-of-flight (MALDI-TOF) mass spectroscopy and RPHPLC at 214 nm. The sequences of the peptides were submitted in international patents, which were published earlier (“*Novel antimicrobial peptides*” (WO2009/124721; priority: 04/2008); granted in EP (validated in CH, DE, ES, FR, GB), US and JP; “*Means and methods for treating bacterial infections*” (PCT/EP2017/053487; priority: 02/2016)).

### 2.2. Stimulation of Mononuclear Cells (MNC) by LPS

Mononuclear cells (MNC) were isolated from heparinized blood samples obtained from healthy donors as described previously [5]. The cells were resuspended in medium (RPMI 1640), and their number was equilibrated at 5 · 10^6^ cells/mL. For stimulation, 200 L MNC (1 · 10^6^ cells) was transferred into each well of a 96-well culture plate. LPS Ra (from S. Minnesota strain R60) and the LPS/peptide ratio mixtures were preincubated for 30 min at 37 °C and added to the cultures at 20 L per well. The cultures were incubated for 4 h at 37 °C with 5% CO_2_. Supernatants were collected after centrifugation of the culture plates for 10 min at 400 g and stored at 20 °C until immunological determination of tumor necrosis factor alpha (TNF-α), carried out with a sandwich enzyme-linked immunosorbent assay (ELISA) using a monoclonal antibody against TNF (clone 6b; Intex AG, Switzerland) and described previously in detail [5]. 

### 2.3. Cytotoxicity Assays and Hemolysis 

Cytotoxicity was assayed by using the chip-based Bionas system with four different cell types. These include human hepatoma cells (HepG2), human colon adenocarcinoma cells (LS-174T), human acute lymphocytic leukemic cells (Jurkat), and human peripheral blood mononuclear cells (hPBMC), provided by Cell Line Services (Piscataway, NJ, USA) and ProBioGen (Berlin, Germany). The cells were seeded on chips, with a density of 200,000 cells/chip for HepG2 and LS-174T, 300,000 cells/chip for Jurkat, and 2 million cells/chip for PBMC.

Furthermore, the MTT assay was performed to determine the metabolic activity of the cells [14]. In this test, the determination of cell viability is based on the reduction of the yellow water-soluble substance 3-(4,5-Dimethylthiazol-2-yl)-2,5-diphenyltetrazoliumbromid (MTT) into a blue-violet water-insoluble formacan. 

### 2.4. Hemolysis Assay 

The activity of the peptides to lyse freshly isolated human erythrocytes was determined in PBS, pH 7.4 at 37 °C. Dilutions were prepared in duplicate in a round bottom microtiter plate. For that, 20 µL erythrocytes (5 · 10^8^ cells/mL) was incubated with 80 µL of a peptide sample at 37 °C for 30 min in a humidified box. The hemolytic activity after incubation was measured by transferring the supernatants into another empty microtiter plate. This plate was read on a microtiter plate reader at 405 nm. Hemolytic activity was expressed as the percentage of released hemoglobin with respect to water/buffer controls (100% release) or controls processed without peptides (0% release).

Red blood cells (RBCs) were obtained from citrated human blood by centrifugation (1500× *g*, 10 min), washed three times with isotonic 20 mM phosphate-NaCl buffer (pH 7.4), and suspended in the same buffer at a concentration equivalent to 5% of the normal hematocrit. Forty-microliter aliquots of this RBC suspension were added to 0.96 mL of peptide dilutions prepared in the same isotonic phosphate solution, incubated at 37 °C for 30 min, and centrifuged (1500× *g*, 10 min). The supernatants were analyzed spectrophotometrically (with absorbance at 543 nm) for hemoglobin, and results were expressed as the percentage released with respect to sonicated controls (100% release) or controls processed without peptides (0% release) [6].

### 2.5. Chip-Based Cellular System (BIONAS) 

A metabolic investigation with four cells and cell lines on the effect of Pep19-2.5 on selected cell parameters was performed. In detail, see Table 1. 

The analysis was performed in the Bionas^®^ 2500 analyzing system by seeding the cells on the chips, with cell numbers of 200,000 for HepG2 cells and LS-174T cells, and 300,000 for Jurkat cells. The suspended cells were embedded on the day of the experiment in agarose and immediately put into the Bionas^®^ 2500 analyzing system. The settling phase (‘RM’) was 5 h. The peptides were added after 5 h for 24 h, then a regeneration phase followed for 4 h (‘RM’), and in the last phase the cells were lysed with 0.2% triton in fresh medium (‘TX’). For the parameters acidification, respiration, and adhesion (impedance), the mean values from two experiments were calculated. The tested concentrations in two independent experimental runs were 0.1, 1, 10, and 100 µg/mL and 10, 20, and 50 µg/mL. 

### 2.6. Modelling of Peptide Structure

The 3D structure of Aspidasept was estimated by calculations with the OPEP method [15,16] using the software Pep-Fold [17]. A previously established 3D pharmacophore model was used to investigate and model potential interactions of the predicted Aspidasept structure with the hERG ion channel as previously described [18]. The training set of the established pharmacophore model had previously been focused on small molecules. To extend the pharmacophore model to peptides, structural similarities between Aspidasept and previously published peptide inhibitors of the hERG channel [19] were investigated and included in the model.

### 2.7. Electrophysiology

Manual patch clamp experiments were performed in the whole cell configuration [20] at room temperature. An EPC 10 patch clamp amplifier and the PatchMaster software (v2x90.5; both HEKA Elektronik, Germany) were used. The patch pipettes were made of borosilicate glass (GB150TF-10; Science Products, Germany) and had a resistance in the range of 2–3 MOhm. The data were analyzed using R (version 3.6.3) [21].

The manual patch clamp recordings on the hERG ion channel were performed with the same stimulation protocol as previously described [22], with an adjusted holding potential of −70 mV. After the establishment of the whole cell, the stimulation protocol was executed every 15 s for a period of 20 min. The application protocol had the following course: 5 min negative control period with an application of 0.1% DMSO, 10 min application of the test compound in one concentration, and 5 min positive control period with an application of E4031 (10 µM).

The extracellular solution for the recordings contained (in mM): 145 NaCl, 4 KCl, 10 glucose, 10 HEPES, 2 CaCl_2_, and 1 MgCl_2_; pH was adjusted to 7.4 by adding NaOH. The osmolarity of the solution was adjusted to 305 mOsmol/kg. The intracellular solution contained (in mM): 120 KCl, 1.7 MgCl_2_, 4 K_2_ATP, 5 CaCl_2_, 10 HEPES, and 10 EGTA; pH was adjusted to 7.2 by adding KOH. The osmolarity was adjusted to 292 mOsmol/kg.

For automated patch clamp experiments, QPatch, a planar patch clamp robot, was used [23]. QPatch experiments were performed as previously described [22,24]. In brief, the hERG ion channel (gene ID 3757) was transfected into CHO cells and a stably expressing cell line was constructed. Patch clamp experiments were performed in the whole-cell voltage clamp mode on a planar patch clamp chip. Successfully clamped cells were stimulated with the following stimulation protocol: From a holding potential of −80 mV, the cells were partially depolarized to −50 mV to test the leak current at this potential where hERG channels are closed. This was followed by a depolarization to +40 mV. At this depolarized potential, the hERG channels alternate between the open and inactivated states. After partially repolarizing to −50 mV, the inactivated subpopulation of hERG channels rapidly recovers from inactivation and switches into the open state, from where the channels slowly close. This results in a large hERG-mediated tail current that is observed in the second partial depolarization phase to −50 mV.

Aspidasept (Pep19-2.5) was applied at nominal concentrations of 1 µg/mL, 10 µg/mL, and 100 µg/mL, and remaining hERG mediated potassium channel currents normalized to initial currents were calculated.

### 2.8. Irwin Test (Neurobehavioral Activity)

In this test, an examination is done whether or not drugs induce behavioural and physiologically relevant effects. Effects were determined for three different doses of Pep19-2.5 after a single i.v. administration versus one dose of a positive (chlorpromazine) control item and a negative control item (saline), respectively, in male Wistar rats using the modified Irwin test. The modified Irwin test was performed under GLP conditions and was comprised of a standard observation battery of tests addressing various parameters of the central nervous, autonomic nervous, peripheral motor, and sensory systems. 

Details of the performed Irwin test are depicted in Figure 1.

The study comprised five treatment groups with six male animals each. The experimental groups and doses are shown in the Table 2.

The following investigations were made at the following time points:Mortality: twice daily during the observation periodClinical observations: on the day of treatment before applicationBody weight development: on the day of delivery and on the day of treatment before applicationModified Irwin test: at the time points pre-test (24 h prior to administration), 1 h, 4 h, 7 h, 24 h, 48 h, and 72 h after application.

Statistical analyses of the data were performed for each group separately. Group mean values (means) with standard deviation (SD) were calculated for body weights, each parameter of the Irwin Test, and each investigation time point.

For Irwin Test parameters, statistical differences between the positive control group and the vehicle control group and between the test item groups and the vehicle control group were evaluated by the Mann–Whitney U-test, using an MS Excel^®^ statistics tool developed and validated in-house; *p* < 0.05 was accepted as the level of significance.

No relevant neurobehavioral effects of Pep19-2.5 were calculated.

### 2.9. Local Lymph Node Assay (LLNA)

The GLP (Good Laboratory practice)-compliant experimental set-up was as follows:Test item concentrations were based on the results of the formulation evaluation and preliminary irritation/toxicity tests according to relevant guidelines (for example, S7A).The maximum attainable test concentration based on solubility was 5% (*w*/*v*) in aqueous 1% Pluronic^®^PE9200 (aqueous 1% Pluronic) using concentration series recommended by relevant guidelinesSince no adverse effect of the test item was observed during the preliminary irritation/toxicity test up to this maximum concentration, Pep 19-2.5 was tested in the main test at 5%, 2.5%, 1%, and 0.5% (*w*/*v*) in aqueous 1% PluronicMale (day 6: 208–259 g; day 1: 244–314 g) Wistar ratsFive animals per group treated with one single i.v. dose of the respective formulation, application volume: 2 mL/kg

### 2.10. Repeated Dose Toxicology in Rats

Continuous intravenous infusion was performed over 14 days with 0.5 mg/kg, 3.2 mg/kg, and 20.0 mg/kg Pep19-2.5 and compared to the vehicle in male and female Wistar rats. Groups comprised 10 animals per gender, with an additional 5 animals per gender for recovery animals in the vehicle and high dose groups. The details of the 14-day repeated dose toxicity study will be published elsewhere.

### 2.11. Fluorescence Resonance Energy Transfer Spectroscopy (FRET) 

The ability of the peptides to intercalate into phospholipids PC and PS was investigated. Phospholipid liposomes from phosphatidylserine (PS), phosphatidylcholine (PC) were doubly labelled with the fluorescent phospholipid dyes N-(7-nitrobenz-2-oxa-1,3- diazol-4yl)-phosphatidyl ethanolamine (NBD-PE) and N-(lissamine rhodamine B sulfonyl)-phosphatidylethanolamine (Rh-PE) (Molecular Probes). Intercalation of unlabeled molecules into the doubly labeled liposomes leads to probe dilution and, with that, to a lower FRET efficiency: the emission intensity of the donor ID increases and that of the acceptor IA decreases (for clarity, only the quotient of the donor and acceptor emission intensity is shown here). In all experiments, the peptide (100 µL of 100 µM) was added to doubly labelled PS or PC phospholipid liposomes at 50 s after equilibration. NBD-PE was excited at 470 nm, and the donor and acceptor fluorescence intensities were monitored at 531 and 593 nm, respectively. The fluorescence signal ID/IA was recorded for another 250 s.

## 3. Results

### 3.1. General In Vitro Toxicology

Data were generated by studying the possible cytotoxic effects of Pep19-2.5. As the most relevant blood cells, mononuclear cells and red blood cells were used. The former was investigated by the MTT test, which examines the metabolic state of the cells (Figure 1), and the latter in the hemolysis assay (Figure 2). As shown, in both test systems there is an increase of cytotoxicity at concentrations above 20 μg/mL. Whereas the metabolic state of the hMNC (human mononuclear cells) decreases dramatically at 100 μg/mL, in the hemolysis assay, 20% lysis are not exceeded even at 100 μg/mL. These data are characteristic for all compounds from the Pep19-2.5 series (Figure 3) and are also valid for the potential metabolites of Pep19-2.5 (Table 3).

### 3.2. Toxicity in a Chip-Based System

The general toxicity of the metabolic behavior of selected cells (PBMCs) and cell lines (HepG2, LS-174T, Jurkat) was analysed using the Bionas^®^ system [25]. In this system, cells were spread from cell suspensions on a chip in agarose, and the parameters determined were the acidification of cells, their respiration, and their adhesion (impedance measurements). As one example, the respiration rate in human colon adenocarcinoma cells is shown, indicating that at concentrations above 20 μg/mL, toxic effects can be observed (Figure 4). In a similar way, the respiration rate in PBMCs was tested, but it increased significantly only at concentrations above 50 μg/mL (Figure 5). Data such as in Figure 4 and Figure 5 indicate that for the different cell types and parameters, significant toxic effects start at Pep19-2.5 concentrations above 30 μg/mL. Overall, the evaluation of the three parameters was conclusive and indicated similar concentrations at which cytotoxic effects are seen. 

### 3.3. Molecular Modeling of Pep19-2.5

The peptide sequence and a representative structural model of Aspidasept, as calculated by Pep-fold [17,26], is presented in Figure 6. Although the structure prediction includes a number of uncertainties due to the great amount of freedom, all possible models show a small number of beta sheets and alpha helices, or even none of these structural elements (Figure 6B).

Predicted Aspidasept structures share no similarity with established pharmacophore models of hERG blockers (Figure 6D) [18]. No successful docking of any of the large Aspidasept structures into the hERG channel pore could be obtained (as assessed by stereo-visual proximity values), which indicates no central pore blockade of the hERG channel by Aspidasept. 

For scorpion toxins, in particular the highly potent hERG blocking peptide BeKm-1, surface interactions between alpha helices and the surface of the hERG channel pore have been reported as the likely sites of interaction and the likely causes of hERG channel blockade [27,28,29]. We have found no corresponding alpha helices that would fit to the hERG channel surface in our predicted models of Aspidasept (Figure 6B,C). In conclusion, the pharmacophore model, the 3D docking model, and the surface calculation of Aspidasept predict no relevant probability of hERG channel blockade by Aspidasept.

It should however be noted that the structures of Aspidasept used here are predicted and calculated by Pep-fold. Therefore, they may contain uncertainties with respect to the individual positions of amino acids that are not readily visible from the graphical representation. The prediction of potential hERG channel interactions could be affected by these uncertainties. Therefore, electrophysiological patch clamp experiments have been performed to further assess any potential effects of Aspidasept on hERG ion channel currents.

### 3.4. Electrophysiology

Manual patch clamp electrophysiology is considered the gold standard for assessing pharmacological effects on ion channel currents. Therefore, the effects of Aspidasept and Pep19-4LF on hERG channel currents were investigated using this method. The sensitivity of the assay was validated with the positive control compound E-4031, which is an experimental class III antiarrhythmic drug that blocks potassium channels of the hERG-type, and stability of the assay was assessed with negative (vehicle) control experiments. No significant effects of the peptides on the hERG-mediated currents could be detected at 1 µg/mL (Aspidasept: 100.1 ± 4.2% (*n* = 3); Pep19-4LF: 97.5 ± 15.0% (*n* = 3)). At higher concentrations, a spontaneous increase in leak currents was observed, indicating unspecific effects of the peptide on the membrane and limiting the performance of the assay. Therefore, no successful experiments by manual patch-clamp could be performed at higher concentrations. 

We transferred our hERG patch clamp assay to the QPatch and validated this with vehicle control and reference compound E4031 experiments. The data show stable measurements (mean current reduction of vehicle control was 4.2 ± 4.3%) in vehicle control experiments and high sensitivity of the assay (mean current reduction of 1 µM positive control E-4031 was 96.8 ± 3%).

On the QPatch, concentrations of 1 µg/mL and 10 µg/mL Aspidasept could be successfully tested on hERG-expressing cells. Figure 7 shows hERG-mediated current traces over time in absence (purple) and presence (green) of 10 µg/mL Aspidasept, indicating only a very minor and not significant current reduction upon application of Aspidasept. No significant effects of the peptide on the hERG-mediated currents could be detected at 1 µg/mL (relative remaining current of 97.2 ± 2.2%, *n* = 4) and at 10 µg/mL (relative remaining current of 95.2 ± 2.5%, *n* = 4) of Aspidasept. 

Upon application of 100 µg/mL (*n* = 6), a spontaneous increase in leak currents was observed that did not allow successful measurements meeting relevant quality criteria, again indicating unspecific membrane effects that limit the performance of the assay. Therefore, no successful experiments could be performed at this concentration. 

Experiments upon application of different concentrations of Pep19-4LF on hERG-expressing cells in the QPatch showed remaining currents of 97.8 ± 5.8% at 0.1 µg/mL (*n* = 3), 99.1 ± 1.7% at 1 µg/mL (*n* = 3), and 94.6% at 10 µg/mL (*n* = 1), corresponding to no significant effect of Pep19-4LF at the concentrations tested.

It has previously been reported that some compounds exhibit a slow effect on hERG channel currents [18]. This is especially relevant for particularly hydrophobic compounds that, in some cases, share large homology with the hERG model pharmacophore and may exhibit slow effects in patch clamp experiments (also for the reason that they may reach the cell at reduced effective concentrations). We have, therefore, extended the number of compound applications and the duration of compound applications according to previously published protocols [22]. Upon application of 10 µg/mL of Aspidasept for up to 20 min, relative remaining currents of 92.2% have been observed, corresponding to no significant current reduction. The data of relative remaining currents upon application of different concentrations of Aspidasept are summarized in Table 4. 

A half-maximal inhibiting concentration (IC_50_ value) for the effect of Aspidasept on the hERG-mediated potassium current could not be calculated, because no significant effect on the hERG channel current at the highest concentration successfully tested (10 µg/mL) could be observed (Figure 8). Thus, the hypothetical IC_50_ value is estimated to be >>10 µg/mL. 

### 3.5. Safety Pharmacology: Neurological Behaviour

Next, we performed safety pharmacology studies to address neurological behaviour (Irwin test) and skin sensitization (local lymph node test) of Pep19-2.5. We determined the neurobehavioral effect of three different doses of Pep19-2.5 after single intravenous administration versus a single dose of a positive (chlorpromazine) and a negative (saline) control in rats (Table 2). 

### 3.6. Irwin Test

The treatment of male Wistar rats with the positive control Chlorpromazine resulted in statistically significant effects on the behavioral profiles, as expected: general, motor activity/coordination, muscle tone, awareness, central excitation and mood, the neurologic profile reflexes, the autonomic profile, and cardiovascular/respiratory system. These results show the power of the present modified Irwin test in detecting multiple neurobehavioral effects.

No effect of Pep19-2.5 was observed in the low dose group (0.02 mg/kg). In the mid dose group (0.2 mg/kg), significant differences to the negative control were noted in rearing, spontaneous locomotor activity and sensitivity to pinching the tail. Significant differences were also observed in the high dose group (2 mg/kg), in terms of sensitivity to pinching the tail and vocalization when touched and hind limb reflex. The observed adverse effects in the mid and high dose group are considered to be moderate compared to the negative control (physiological saline). Dose-dependent effects of Pep19-2.5 were suggested for the parameters sensitivity to pinching the tail (behavioral profile: central excitation), vocalization when touched (behavioral profile: mood), and hind limb reflex (neurologic profile: reflexes). At the one hour observation time point, body temperature and spontaneous locomotor activity were slightly but not significantly reduced, with a slight dose dependence. No other dose-dependent effects were observed (weight, clinical observations).

#### Statistical Analysis

The statistical analysis showed that the pre-values were within the range considered normal for this species; however, response to the approaching finger was reduced (median −2 in the control groups and −1.0 in the low dose test item group) in animals of the negative control group, the positive control group, and 3/6 animals of the low dose (0.02 mg/kg b.w.) test item group. This resulted in marked and statistically significant differences between the negative control group and the mid and high dose (0.2 and 2 mg/kg b.w.) test item groups (median 0.0 in both test item groups). Other minor differences between the groups were considered to be of no biological relevance.

Upon treatment with the positive control Chlorpromazine, a reduced rearing 1 h after injection (*p* < 0.05) was observed. In addition, an effect on the muscle tone profile (hind limb tone and grip strength) 1 h after treatment was observed (*p* < 0.05). Chlorpromazine also had several significant effects on the behavioral profiles awareness, central excitation, and mood 1 h after treatment (*p* < 0.005 and *p* < 0.05). A significant influence on the neurologic profile (reflexes) and the autonomic profile (hind limb reflex and eyes opening; *p* < 0.005), as well as on the cardiovascular/respiratory system (cyanosis; *p* < 0.05), were observed upon injection of Chlorpromazine. 

The observed effects on the profiles listed above were reversible and are considered to be related to the treatment with Chlorpromazine. All further values of the different profiles were within the normal range for the species, and no biologically relevant differences were observed. These results show the power of the present modified Irwin test in detecting multiple neurobehavioral effects.

*Application of the test item* Pep19-2.5 (at 0.02, 0.2 and 2 mg/kg b.w.):

In the behavioral profile **motor activity/coordination**, some animals of the dose test item group showed a tendency toward reduced spontaneous locomotor activity 1 h after the treatment. This was observed for 3 animals of the low (0.02 mg/kg b.w.; median −0.5), 4 animals of the mid (0.2 mg/kg b.w.; median −1.0), and 5 animals of the high (2 mg/kg b.w.; median −1.0 vs. 0.0 in the negative control group) dose test item groups. However, the observed differences were not statistically significant. Animals treated with the mid dose Pep19-2.5 (0.2 mg/kg b.w.) showed more spontaneous locomotor activity 48 and 72 h after treatment than the animals of the saline-treated (negative control) group (median 0.5 vs. −1.0). This difference was statistically significant at the 72 h observation time point (*p* < 0.05). Since no effect was visible in the high dose test item group, the finding is not considered to be test item related. Additionally, the animals of the mid dose group reared more often than the animals treated with the negative control (saline) at time points 48 and 72 h after treatment. The differences were slight to moderate and were statistically significant (median 1.0 vs. −1.0, *p* < 0.05 at 48 h; 1.0 vs. −1.0, *p* < 0.01 at 72 h). A dose dependency in the parameter rearing was not visible, and no significant differences occurred at any other time point. Therefore, the effects are not considered to be related to the test item. 

The value of the hind limb tone in the behavioral profile **muscle tone** was slightly but significantly higher in the low dose test item group (median 2.0) when compared to the negative control group (median 1.0) 72 h after treatment. However, no dose dependency was recorded, and values are considered to be within the normal range of variation. Therefore, the difference is not considered to be test item related nor of biological relevance.

In the behavioral profile **awareness**, the value of the finger approach was markedly and significantly higher in the mid dose test item group 72 h (median 2.0) after treatment than in the saline control group (median 0.0, *p* < 0.05). As previously described, already in the pre-test, animals of the test item groups reacted differently to the approaching finger than the animals of the negative control group. Therefore, the difference at the 72 h time point is not considered to be test item related. The value head touch was slightly and significantly higher in the group treated with the mid dose of Pep19-2.5 than in the saline control group 7 h after the treatment (median 0.0 vs. −1.0 in the saline control group, *p* < 0.01). No dose dependency was observed, and the value was within the normal range of variation. The change in value is therefore not considered to be an effect of test item application.

The parameter **positional passivity** in the low dose test item group significantly deviated from the saline-treated group 1 h after treatment (median 0.0 vs. 1.0, *p* < 0.05). No dose dependency was visible for this parameter, and the values are in the normal range of variation. The difference in positional passivity is not considered to be related to the administration of the test item.

In the behavioral profile **central excitation,** the sensitivity to pinching of the tail was moderately reduced or absent in 2 animals of the low dose, in 4 animals of the mid dose, and in 5 animals of the high dose test item group 1 h after treatment. The value in the high dose test item group was significantly lower compared to the negative control group (median high dose −1.5 vs. 0.0 in the negative control group; *p* < 0.01). At the 7 h observation time point, a significant difference between the high dose test item group and the saline-treated control group was visible as well (median −1.5 vs. 0.0, *p* < 0.05). A dose dependency was visible for this parameter especially at the 1 h observation time point (median: low dose 0.0, 2 animals with reduced reactivity; mid dose −1.5, 4 animals with reduced reactivity; high dose −1.5, 5 animals with reduced reactivity). This tendency toward reduced sensitivity to pinching of the tail 1 and 7 h after treatment is considered to be related to the test item. At all later time points, no relevant differences were visible between the saline and the test item groups in the central excitation profile.

In the behavioral profile **mood,** significantly more animals of the high dose test item group showed vocalization when touched than animals of the saline-treated control group 1 and 24 h after the treatment (median high dose test item group 1.0 vs. 0.0 in the control group, *p* < 0.05). Animals of the high dose test item group showed a tendency of increased vocalization over the entire observation period. At the 1 h observation time point, a dose dependency was visible for the parameter vocalization when touched. The differences in vocalization are considered to be related to the administration of the test item. No other differences between the test item groups and the saline control group occurred in the mood profile.

### 3.7. Local Lymph Node Assay (LLNA)

The GLP-compliant experimental set-up was as described in Materials & Methods and is shown in Figure 2 and Table 5. The results for Pep19-2.5 revealed no skin sensitization potential either at the maximum attainable concentration (5%) or at concentrations of 2.5%, 1%, and 0.5% (*w*/*v*) formulated in 1% Pluronic aqueous solution. 

Application of substance on external surface of each ear (25 µL/ear) for 3 consecutive daysOn day 6, intravenous injection of tritiated methyl thymidine (^3^HTdR) in tail vein, and after about 6 h, sacrification and preparation of lymph node cellCell proliferation in lymph node measured by incorporation of ^3^HTdR

### 3.8. Repeated Dose Toxicology in Rats 

Continuous intravenous infusion was performed over 14 days with 0.5 mg/kg, 3.2 mg/kg and 20.0 mg/kg Pep19-2.5 and compared to the vehicle in male and female Wistar rats. Groups comprised 10 animals per gender, with an additional 5 animals per gender for recovery animals in the vehicle and high dose groups. Based on the parameters determined, the following results were found:

Permanent catheterization strongly deteriorated all animals, including those of the control group. It is assumed that the corresponding catheter effects enhanced possible adverse effects of the test item in the high dose group. This assumption was ascertained by the recovery of all surviving animals, as well as the absence of any histopathological systemic toxicity. One male of the high dose group died (presumably) due to a test item-related effect. Two additional animals of the same group had to be euthanized, but it is unclear whether this was due to the test item or catherization. In the high dose group, single animals demonstrated test item-related behavioural findings (increased activity, muscle tremor, increased incidence of animals showing signs of discomfort, such as chromorhinorrhea or piloerection). Only limited recovery was noted.

A local irritating effect leading to increased incidences of findings associated with the injection site was noted for the 20 mg/kg dose group. This was confirmed by macroscopic findings. An individual variation regarding food consumption was noted in animals of all dose groups. No statistically significant effect on the body weight gain was noted. Ophthalmology, parameters of clinical chemistry, urine, and organ weights were not affected by the treatment. 

Coagulation parameters (prothrombin time (PT), activated partial thromboplastin time (aPTT), and thrombin time (TT)) were not affected by the treatment with Pep19-2.5 at the end of the infusion period in both genders compared to the vehicle group, except for a slight and dose-dependent shortened mean prothrombin time in females of all dose groups. In detail, mean values ± standard deviation were: For male Wistar rats, aPTT values were 19.8 ± 2.6 s (*n* = 9) in negative control, and under treatment of Pep19-2.5: 19.9 ± 1.7 s (*n* = 9) for 0.5 mg/kg/day, 18.6 ± 1.8 s (*n* = 10) for 3.2 mg/kg/day, 19.1 ± 1.5 (*n* = 7) for 20 mg/kg/day. PT values were 20.2 ± 2 s (*n* = 9, negative control); 20.0 ± 1.1 s (*n* = 9, 0.5 mg/kg/day); 21.5 ± 1.3 s (*n* = 10, 3.2 mg/kg/day); 20.9 ± 1.3 s (*n* = 7, 0.5 mg/kg/day). TT values were: 24.8 ± 1.1 s (*n* = 9; negative control); 25.3 ± 1.0 s (*n* = 9; 0,5 mg/kg/day); 25.0 ± 0.9 s (*n* = 10; 3.2 mg/kg/day); 25.0 ± 0.8 s (*n* = 7; 20.0 mg/kg/day). For female Wistar rats, aPTT values were 18.0 ± 1.4 s (*n* = 10; negative control); 19.9 ± 2.2 s (*n* = 10; 0.5 mg/kg/day); 17.9 ± 2.3 (*n* = 10; 3.2 mg/kg/day); 18.7 ± 2.0 (*n* = 10; 20 mg/kg/day). PT values were (Asterisks indicate significant differences to negative control with * *p* < 0.05 and ** *p* < 0.01): 22.1 ± 1.5 s (*n* = 10, negative control); 20.3 ± 2.5 s (*n*= 10; 0.5 mg/kg/day); 20.1* ± 1.9 s (*n* = 10; 3.2 mg/kg/day); 19.9** ± 1.4 (*n* = 10; 20.0 mg/kg/day). TT values were 25.3 ± 1.1 s (*n* = 10, negative control); 25.7 ± 1.0 s (*n* = 10; 0.5 mg/kg/day); 25.4 ± 1.0 s (*n* = 10; 3.2 mg/kg/day); 25.8 ± 0.7 s (*n* = 10; 20.0 mg/kg/day).

In single animals, gelatinous plasma was noted. The values of these plasma samples had to be excluded in the mean value calculation and statistics. A reason for the gelatinous plasma samples is unclear, but a test item-related effect was excluded as a vehicle-treated animal was affected as well. At the end of the treatment period, a slightly shorter mean prothrombin time was noted in all female dose groups dose dependently, with statistical significance in the mid and high dose groups.

Reticulocytes in the high dose group were slightly reduced in both genders, but showed a complete recovery.

Based on the effects determined, the dose of 0.5 mg/kg∙d and 3.2 mg/kg∙d were assumed to be within the No-Observed-Adverse-Effect-Level (NOAEL). The 20 mg/kg dose was within the maximum tolerated dose (MTD), as adverse reactions were noted. 

In a similar way, Pep19-4LF was analysed. However, due to problems with permanent catheterization in the former study, the animals were treated intravenously with the test item for 30 min each day. 

In the dose escalation phase, there was no lethality at any of the tested dose level following single intravenous administration to females. No clinical signs were noted at dose levels <20 mg/kg. After treatment of dose 20 mg/kg, water consumption was increased. Minimal body weight loss was observed following treatment with doses of ≥1 mg/kg. The body weight normalized the next day at 1 and 5 mg/kg, while at 20 mg/kg between days 3–4. 

In the repeated dose phase, no systemic clinical signs were noted. The only clinical sign was observed locally at the site of administration toward the end of the treatment period (between days 2–5) and was livid (purple) to black discoloration of the distal part of the tail. 

Minimal body weight loss was observed following the first day of administration to the end of in life phase. Overall body weight loss of approximately 10% was observed in males and 2% in females (Figure 9). At necropsy (on Day 6), no remarkable observation was made, with the exception of the local finding of administration sites (black discoloration of tail ends).

The amount of available data does not allow exact determination of NOAEL and MTD. However, the comparison with Pep19-2.5 shows significantly lower toxic effects for Pep19-4LF. 

### 3.9. Origin of the Toxicology of the Pep19-2.5 Series

Though the details of the observed side effects in the different biological systems may be different, one basic toxicological property may be the possibility that polypeptides, in particular those from the Aspidasept series having an amphiphilic character, tend to incorporate into target cell membranes. To test this in experiments with Förster resonance energy transfer (FRET) spectroscopy, a possible intercalation of the peptides into membranes built from phospatidylcholine (lecithin, PC) or phospatidylserine (PS), which are the main zwitterionic and negatively charged constituents of natural membranes, respectively, was tested. The intercalation of labeled molecules into the doubly labeled liposomes leads to probe dilution and, with that, to a lower FRET efficiency; therefore, the emission intensity of the donor I_D_ increases and that of the acceptor I_A_ decreases (Figure 10). 

In summary, there is a strong intercalation of the peptide into PS membranes and a lesser intercalation, but still significant, into PC membranes. This means that the intercalated peptides may lead to strong cell membrane disturbances, which should be responsible for the observed toxic effects in the biological systems. The details of this mechanism and proposed working hypothesis, however, have still to be evaluated. 

## 4. Discussion and Conclusions

A broad analysis of the toxicological behavior of Pep19-2.5 and some derivatives was performed in various in vitro systems, as well as in animal models, according to ICH M3, ICH S7A, and ICH S7B. 

The data presented here for mononuclear cells, PBMC, and erythrocytes (hemolysis) (Figure 1, Figure 2 and Figure 5) are consistent with the animal data. For example, when recalculating the NOAEL of rats of 3.2 mg/kg∙d to volume concentrations in blood (the relevant quantity), one arrives (with an estimated 6 L blood) at approximately 0.5 μg/mL, which is even higher than in most test systems. Scheduled therapeutic concentrations in sepsis patients would lie below any toxic effects, i.e., in the range of 30 to 50 μg/kg body weight or to 40 to 60 ng/mL blood (corresponding to approximately 1/10th of the calculated NOAEL). Therefore, the “therapeutic window” is high enough, i.e., therapeutic treatments of sepsis patients can be done most readily without severe side effects. 

It was reported that low toxicity was found also for hepatoma (Huh7.5.1) cells and HepaRG cells up to concentrations larger than 40 μg/mL [30]. Further investigations with skin and soft tissue cells indicate a similar behavior [31]. An increase in toxic effects was found for human keratinocytes, starting at 30 μg/mL. For monocyte-derived dendritic cells and Langerhans cells, cytotoxic effects start at concentrations >> 30 μg/mL. 

In the present study, different tests systems (cell-based using dyes and chip-based systems; Figure 4 and Figure 5) have been employed for the assessment of potential effects. Comparing these data indicate potential significant toxic effects of Pep19-2.5 at concentrations above 30 µg/mL. Moreover, for different cell-based systems, unspecific effects have been observed at high concentrations. These might be related to the intercalation of the peptides into the lipid membrane (Figure 8). The details of these mechanisms, however, have to be elucidated in further studies. 

Interestingly, selected part structures of Pep19-2.5, which may be metabolites when applicated in vivo, share the same behaviour with their parent compound in the hemolysis assay (Figure 3). Thus, the metabolites do not represent a safety problem in clinical use, which is an important finding with respect to permission by regulatory authorities. 

Inhibition of the potassium current flowing through the cardiac hERG ion channel (Kv11.1) has been correlated to life threatening torsades de pointes cardiac arrhythmia [12]. Effects on hERG channel currents have been observed for a number of drugs, including antihistamines, antipsychotics, and many other drugs from nearly all therapeutic areas and chemical classes, and have become a serious safety concern in drug discovery and development. Inhibition of hERG channel currents have also been reported for antibiotics, in particular for macrolides, raising concerns about the cardiac safety of antibiotics [32,33,34,35,36]. Blockade of the hERG ion channel has been correlated to occupation of the relatively large hydrophobic cavity inside the channel, and pharmacophore models have been established that predict blockade of the channel for certain structural classes of small molecules with high predictive power [18,37]. In addition to the blockade by small molecules, peptides have been identified that affect the hERG ion channel’s conductivity. In particular, scorpion toxins, e.g., BeKm-1, block hERG channels with high affinity, and residues critical for BeKm-1 binding to the hERG channel have been identified in the alpha-helix and the following loop, indicating a unique localization of BeKm-1′s interaction surface and a specific interaction with the hERG channel [19].

To address potential cardiac safety concerns of Aspidasept, we have performed a thorough analysis of potential effects on hERG channel currents using modelling, manual patch clamp, and automated patch clamp electrophysiology using validated Qpatch technology in transfected cells. The in vivo plasma concentration of Aspidasept applicated in sepsis patients continuously injected can be estimated to 200 ng/mL. While the complete absence of significant hERG effects would be most desirable in drugs of non-cardiac therapeutic application, it has previously been discussed that compounds exhibiting a factor of 100 between the plasma concentration and the measured IC_50_ of effects on the hERG channel current may offer a sufficient safety margin and degree of safety concerning cardiac hERG effects [11].

The effects of Aspidasept on hERG-mediated potassium channel currents were investigated by theoretical molecular docking, and experimentally by manual and by planar patch-clamp electrophysiology. No effects of Aspidasept on hERG-mediated potassium channel currents were observed. By manual patch clamp, concentrations of up to 1 µg/mL were investigated. By planar patch clamp, concentrations of up to 10 µg/mL were successfully investigated. At higher concentrations, unspecific effects limited the performance of the assay and no successful experiments could be performed. Such unspecific effects might be attributed to lipid-effecting properties of compounds at high concentrations that can lead to a deterioration of the lipid membrane and, therefore, a loss of the measurement seal. Consequently, our observations of unspecific membrane effects from the patch clamp experiments are consistent with the FRET data presented here, which indicate a strong intercalation of the peptide into PS membranes and a lesser, but still significant, intercalation into PC membranes. Interestingly, in planar patch experiments, higher concentrations than in manual patch-clamp experiments were successfully investigated. It is likely that the larger surfaces in the planar experiment stabilize the planar patch clamp experiments, and thus allow for the possibility to investigate higher concentrations of Aspidasept by planar patch clamp than by manual patch clamp.

In conclusion, in vitro cell-based and in vivo data of Aspidasept structures are consistent and suggest a safe therapeutic window of application of Aspidasept. At concentrations of 20 to 30 µg/mL, side effects are observed in cellular systems and the NOAEL (No Observed Adverse Effect Level) and MTD (Maximum Tolerated Dose) is estimated at more than 3 mg/d∙kg and 20 mg/d∙kg, respectively. In rats in repeated toxicology tests with continuous intravenous application, a slight neurobehavioral effect at the highest dose (2 mg/kg, formulated in 0.9% saline), no effect on skin sensitization up to 5% formulated in pluronic aqueous solutions, and no effect on the hERG channel at concentrations well above 10 µg/mL were observed. These data together indicate an acceptable level of toxicity, and scheduled therapeutic concentrations in sepsis patients (in the range of 30 to 50 μg/kg body weight or to 40 to 60 ng/mL blood) would lie below these potential toxic levels.

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
