# Peer review of "Toxicological and Safety Pharmacological Profiling of the Anti-Infective and Anti-Inflammatory Peptide Pep19-2.5"

_microorganisms, 2022, doi:10.3390/microorganisms10122412_

Round 1

Reviewer 1 Report

The authors present a preliminary toxicology study of a novel series of LPS binding peptides that have anti-inflammatory and antimicrobial activity (previous work).  A number of in vitro and in vivo studies were done and suggest that toxicity is low at concentrations that would be used clinically.  The information is necessary for further development of these peptides.  There are some weaknesses that need to be addressed.

1. There is a lack of statistical analysis of the data.

2.  A bit more detail needs to be included regarding the Irwin test so readers don't need to go to the literature.

3.  They don't specify how many animals were used in the Irwin test.

4.  The paragraph in the discussion that begins with "Taking all the in vitro data together..."  is confusing.  The point they are making is not clear so revision is needed.

5.  For presentation purposes it appears they used different programs to generate the figures.  The formats should be standardized.  For example compare Figures 1 and 2.

Author Response

We thank the reviewer for their careful and critical assessment of our manuscript, and for the constructive comments. We agree to the reviewers comments; please see attachment.

Reviewer 2 Report

This paper by Möller et.al, describes investigations regarding toxicological and safety pharmacological profile of the peptides pep19-2.5 and pep19-4LF. These anti-inflammatory agents are promising candidates for treatment of sepsis and septic shock.

The authors show that above concentrations of 20 µg/ml side effects are observed in cellular systems. The MTD (Maxiumum Tolerated Dose) was calculated at 20 mg/d∙kg. With this dose 30 % of the animals died or has to be killed due to adverse effects (with pep19-2.5), is this an acceptable level of toxicity? The authors calculated the NOAEL with 3.2 mg/kg∙d. However, how was then the therapeutic dose for sepsis patients (30 - 50 µg/kg, 400-600 ng/ml in blood, see discussion) calculated? This information has to be included into the discussion.

For the toxicology study more data have to be shown, which are described in the manuscript, but not shown e.g. body weight. Moreover, it has been described that pep19-2.5 interferes with coagulation in vitro, therefore data for in vivo coagulation must be shown. Which parameters were determined for coagulation? In view of the present data, this is particularly important because in the in vivo experiments with pep19-4LF the authors described that purple to black discoloration of the distal part of the tail were observed, which might be caused by a disturbed coagulation.

There is a problem with the tables in the manuscript. There are table 1,2,3 then table 2 again, and finally table 8?

Format of the figures is necessary, Fig. 4 and 5 should be the same size, figure 9 is not readable.

I miss the ethical approval for animal experiments.

Author Response

(The authors gave the same response as above.)

Round 2

Reviewer 2 Report

The table legends and numbering are still out of order (there are 6 tables in the manuscript), the table legend must be above the table.

Figure 10 contains two figures that are not identified in the figure legend and they are also unformated

There is still no statement about the ethical approval for the animal study in the manuscript (eg. the project identification code)?

Author Response

We thank the reviewer again for carefully and critically reviewing the revised version of the manuscript, and for the critical comments. 

> "The table legends and numbering are still out of order (there are 6 tables in the manuscript), the table legend must be above the table."

> "Figure 10 contains two figures that are not identified in the figure legend and they are also unformated"

We had previously redone some of the figures (including figure 10) and one table, and had deleted the previous versions of the figures and table using the “track changes” function in MS Word. It seems that, unfortunately, in the pdf version that was provided to the reviewer, the deleted figures were unfortunately not clearly identified as deleted. 

To avoid further confusion, we have now fully deleted the old versions of the figures and tables. We have also checked the table and figure numbering again and checked that the table legends are put above the tables.

> "There is still no statement about the ethical approval for the animal study in the manuscript (eg. the project identification code)?"

The animal studies were approved by the Government of Upper Bavaria and were conducted under the registration number AZ 55.2-1-54-2532.2-13-12 (14 days repeated dose toxicology; Study ID 5414, date of registration May 17th, 2013; date of approval: June 3rd, 2013) and AZ 55.2-1-54-2532.2-9-07 (Irwin Test; Study ID 4554, date of registration Nov 19th, 2013; date of approval: Dec 4th, 2013). This has been added to the manuscript.